evolution, cognition

*Psittaciformes*, longevity, cognitive evolution, Bayesian structural equation model, cognitive buffer hypothesis, expensive brain hypothesis

**Author for correspondence:**
Simeon Q. Smeele
e-mail: ssmeele@ab.mpg.de

†Joint senior authors.

# Coevolution of relative brain size and life expectancy in parrots

Simeon Q. Smeele[1,2,3,5], Dalia A. Conde[5,6,7], Annette Baudisch[5], Simon Bruslund[8,9], Andrew Iwaniuk[10], Johanna Staerk[5,6,7], Timothy F. Wright[11], Anna M. Young[12], Mary Brooke McElreath[1,2,†] and Lucy Aplin[1,4,13,†]

[1]Cognitive and Cultural Ecology Research Group, Max Planck Institute of Animal Behavior, Radolfzell, Germany
[2]Department of Human Behavior, Ecology and Culture, Max Planck Institute for Evolutionary Anthropology, Leipzig, Germany
[3]Department of Biology, and [4]Centre for the Advanced Study of Collective Behaviour, University of Konstanz, Konstanz, Germany
[5]Interdisciplinary Centre on Population Dynamics, University of Southern Denmark, Odense, Denmark
[6]Department of Biology, University of Southern Denmark, Odense, Denmark
[7]Species360 Conservation Science Alliance, Bloomington, IN, USA
[8]Vogelpark Marlow gGmbH, Marlow, Germany
[9]Parrot Taxon Advisory Group, European Association of Zoos and Aquaria, Amsterdam, The Netherlands
[10]Department of Neuroscience, University of Lethbridge, Lethbridge, Canada
[11]Biology Department, New Mexico State University, Las Cruces, NM, USA
[12]The Living Desert Zoo and GardensPalm Desert, Palm Desert, CA, USA
[13]Division of Ecology and Evolution, Research School of Biology, The Australian National University, Canberra, Australia

 SQS, 0000-0003-1001-6615; DAC, 0000-0002-7923-8163; AB, 0000-0002-4202-089X; SB, 0000-0003-4701-1754; AI, 0000-0001-9273-3655; JS, 0000-0001-6965-019X; TFW, 0000-0003-2859-5360; AMY, 0000-0003-3321-1878; MBM, 0000-0003-3206-5485; LA, 0000-0001-5367-826X

Previous studies have demonstrated a correlation between longevity and brain size in a variety of taxa. Little research has been devoted to understanding this link in parrots; yet parrots are well-known for both their exceptionally long lives and cognitive complexity. We employed a large-scale comparative analysis that investigated the influence of brain size and life-history variables on longevity in parrots. Specifically, we addressed two hypotheses for evolutionary drivers of longevity: the *cognitive buffer hypothesis*, which proposes that increased cognitive abilities enable longer lifespans, and the *expensive brain hypothesis*, which holds that increases in lifespan are caused by prolonged developmental time of, and increased parental investment in, large-brained offspring. We estimated life expectancy from detailed zoo records for 133 818 individuals across 244 parrot species. Using a principled Bayesian approach that addresses data uncertainty and imputation of missing values, we found a consistent correlation between relative brain size and life expectancy in parrots. This correlation was best explained by a direct effect of relative brain size. Notably, we found no effects of developmental time, clutch size or age at first reproduction. Our results suggest that selection for enhanced cognitive abilities in parrots has in turn promoted longer lifespans.

## 1. Introduction

Evolutionary theories of ageing predict the inevitability of senescence in most iteroparous multicellular organisms [1–4]. However, recent studies have highlighted the diversity of patterns and timing in which different taxa experience senescence, revealing species-specific patterns of longevity linked with

allometry and life-history variables [5,6]. Generally, larger bodied species tend to live longer [7], but longevity is also associated with other variables such as diet, latitude and sociality [8,9]. Perhaps of most recent interest, brain size has been correlated with longevity across diverse taxa ranging from amphibians [10] to primates [11]. While some studies have proposed a negative relationship between brain size and longevity, suggesting a trade-off between the energetic costs into larger brains and investments in defences against ageing (e.g. [12]), the large majority of studies have suggested a positive effect of larger brain sizes on longevity [10,11,13–17]. However, the causal pathways for this relationship between brain size and longevity are not yet well established.

Three non-mutually exclusive hypotheses have been proposed to explain the correlated evolution of larger brains and longer lifespans. First, the *cognitive buffer hypothesis* posits that increased cognitive flexibility enabled by a relatively larger brain allows species to solve problems that would otherwise increase their extrinsic mortality, hence allowing for increased longevity [15]. Second, the *expensive brain hypothesis* argues that there is an indirect association between brains and longevity, with an investment in expensive brain tissue slowing down the pace of life through increased developmental time and increased parental investment per offspring [18]. Third, the *delayed benefits hypothesis* extends the *expensive brain hypothesis* and reverses the directionality of its argument, positing that a shift to a higher quality, skill-based diet lowered adult mortality rates and supported a longer juvenile period that facilitated inter-generational skill transmission. This extended development in turn allows for investment in brain growth that further promotes skill-based foraging niches. In other words, long-lived, extractive foraging species evolve larger brains because they can benefit most from learning [17]. Previous work in mammals, amphibian and birds has found mixed support for all three hypotheses [13,16]. For example, Isler *et al.* [18] showed that larger brained, monotokous (single offspring per reproduction), precocial mammals had longer developmental periods. This longer developmental period led to a prolonged lifespan; in other words, the effect of brain size on longevity was indirect. By contrast, Jiménez-Ortega *et al.* [14] showed both a direct and an indirect effect of absolute brain size on lifespan in birds, with larger brained species also living longer independently from their developmental period.

Parrots (Psittaciformes) are famous for both their long lives and complex cognition [19,20], with lifespans and relative brain size on par with primates [21]. Indeed, recent studies on the genetics of longevity and cognition in parrots have revealed positive selection on lifespan-prolonging genes, as well as genes related to increased cognitive abilities and cell repair [22–24]. Parrots are also morphologically and ecologically diverse, with an extensive global distribution of almost 400 species, ranging in size from adult yellow-capped pygmy parrots (*Micropsitta keiensis*, 12 g) to kakapo (*Strigops habroptilus*, 3000 g) [25]. In the first comparative study to examine longevity in parrots, Munshi-South & Wilkinson [19] used maximum longevity records from 162 species and found that both diet and communal roosting were correlated with longevity, with granivorous and communal roosting species living the longest on average. While not considering longevity, the potential drivers of the evolution of brain size in Neotropical parrots were explored in Schuck-Paim *et al.* [26], finding that brain size is associated with environmental and seasonal

variability. Finally, highlighting the importance of life-history variation, Young *et al.* [27] found that longer lived parrots were more likely to be threatened. To date, however, little research effort has been invested in understanding the link between longevity and brain size in parrots.

One of the greatest challenges for comparative life-history studies is sourcing good quality data [28]. For instance, the above studies all depended on maximum (or median) recorded lifespan, many used regressions on residuals (e.g. Gonzáles-Lagos *et al.* [29]) and some only included absolute brain size (e.g. Jiménez-Ortega *et al.* [14]). Maximum recorded lifespan can be a problematic measure because it represents the longest-lived known individual and is therefore highly sensitive to sample size. Making matters worse, how much sample size influences results depends on the pattern of age-related mortality itself [30]. For species where most individuals die around the same age, smaller samples are more likely to approximate maximum longevity than in species with many extreme ages of death. Therefore, a measure that accounts for all information available is preferable to a single-point measure. Life expectancy is one such a measure and has been found to be the most appropriate measure of pace of life [31]. It calculates the average age at death based on information across the full age range and therefore takes into account all available information. While life expectancy can be sensitive to both intrinsic and extrinsic sources of mortality, the use of captive records allows the removal of extrinsic sources of mortality as much as possible, thereby focusing on senescence. Yet even when using captive data, other variables and shared evolutionary history create confounds that need to be addressed within a multivariate framework. A principled way to decide which covariates to include is the use of directed acyclical graphs (DAGs) [32,33]. Based on a specific hypothesis, a DAG represents all potential causal paths in the system by arrows. Conditional on the DAG being true, the back-door criterion informs which variables should be included and which should not be included [34]. We additionally controlled for variables that only influence life expectancy to improve accuracy of the model estimates.

Here, we present a phylogenetic comparative analysis focused on brain size and its effects on longevity in parrots. First, we estimate life expectancy from Species360's zoological information management system (ZIMS) with records of 133 818 individuals across 244 parrot species. We then test for a correlation between life expectancy and relative brain size after removing the effect of covariates. Third, we used a DAG to distinguish between two possible pathways for this correlation. The *cognitive buffer hypothesis* predicts a direct effect of relative brain size on life expectancy, with larger brained species living longer [15], while the *expensive brain hypothesis* predicts that the effect of brain size on life expectancy is indirect, emerging from increased developmental time and parental investment per offspring [18]. In this case, we expect that any relationship between brain size and life expectancy will be reduced when also including parental investment (clutch size) and developmental time in the model. While the *delayed benefits hypothesis* would also predict a direct relationship between relative brain size and longevity [17], it would argue for strong effect of diet, as well as reversed directionality (extended longevity leads to larger brain sizes). While we included diet in our models, our analysis focused explicitly on how brain size could affect longevity,

and so we did not fully explore this hypothesis. Overall, our study demonstrates a robust methodology for comparative life-history analysis using a comprehensive measure of life expectancy in a Bayesian statistical framework. Moreover, it provides, to our knowledge, the most comprehensive analysis of longevity in *Psittaciformes* to date and contributes to a broader understanding of this understudied group.

# 2. Material and methods

## (a) Estimating life expectancy

We obtained data on birth and death dates from Species360's ZIMS. After cleaning (see the electronic supplementary material, methods) we included records for 133 818 individuals across 244 species. To estimate life expectancy, we implemented Bayesian survival trajectory analysis (BaSTA, [35]), which allowed us to make inferences on age-specific survival based on census data when ages of some individuals are unknown. The model, implemented in R [36], uses a Markov chain Monte Carlo algorithm with Metropolis-Hastings sampling of mortality parameters and latent times of birth. Here, we used a Siler hazard model [37] for each species, given by

$$\mu(x) = \exp[a_0 - a_1 x] + c + \exp[b_0 + b_1 x],$$

where $a_1, c, b_1 > 0$ and $a_0, b_0 \in (-\infty, \infty)$. These five parameters can fit infant and juvenile mortality (controlled by $a_0$ and $a_1$), age independent (adult) mortality ($c$) as well as senescent mortality (controlled by $b_0$ for initial mortality and $b_1$ for the rate of ageing). Cumulative survival can be calculated as

$$S(x) = \exp\left[-\int_0^x \mu(t)\,dt\right].$$

Life expectancy at birth is calculated as

$$e_0 = \int_0^\infty S(x)\,dx.$$

We used the Gelman–Rubin statistic (Rhat, [38]) to determine if models converged and visually assessed the traces and model goodness of fit. When models did not converge, they were rerun with longer burn-in and more iterations. If models clearly did not fit the data, the results were excluded. This was the case for 27 of 244 species. In most cases this was owing to issues with data quality (e.g. when the number of individuals without a recorded date of death was too high).

## (b) Life-history covariates

We collected body mass data from ZIMS. Additional body mass measurements were included from the literature if no captive records were available for a species [28]. We then used a Bayesian multi-level model to extract species-level averages and standard errors (see the electronic supplementary material, methods). Brain mass was collected by A.I., from Iwaniuk *et al.* [39], from Schuck-Paim *et al.* [26] and from Ksepka *et al.* [40], and similarly to body size, we fitted a Bayesian multi-level model to extract species-level averages and standard errors. We also collected data for six additional potential explanatory variables, based on previously proposed causal relationships with life expectancy: diet (estimated protein content of main food items) [19], insularity (whether a species includes a continental range or not) [19], maximum latitudinal range (as a proxy for environmental variability) [9], clutch size [41], developmental time (from the start of incubation until fledging) and age of first possible reproduction (AFR) [18]. Diet, insularity, maximum latitude range, clutch size and developmental time were collected from the literature.

When data were not freely available, we collected estimates directly from experts (see the electronic supplementary material, methods for details). Finally, AFR is unknown for the large majority of parrot species. We therefore estimated it directly from the distribution of first breeding records in ZIMS, using the 5% percentile. To control for possible issues arising from low sample sizes, we restricted this analysis to species with at least 30 breeding individuals.

We used a DAG (figure 1) to decide how to incorporate variables in the statistical models, accounting for their influences on each other in proposed causal pathways. It is important to note that evolutionary time is not included explicitly in the DAG; thus, arrows can potentially go in both directions, representing evolutionary feedbacks. However, in our view, it represents the most principled representation of the potential causal relationships for evolution of longevity in parrots, based on available data and current knowledge. Although not depicted in the DAG, phylogenetic covariance was assumed to influence all variables and was included in all analyses using the L2-norm (which calculates the covariance between two species based on a maximum possible covariance and the squared distance between the two species) and the phylogenetic tree from Burgio *et al.* [42].

## (c) Statistical analysis

To test for a correlation between life expectancy and relative brain size, we first constructed a Bayesian structural equation model (model 1) with life expectancy as the main variable to be explained by relative brain size and four other potential covariates. We included a total of 360 species for which at least one variable was known. The structure of this first model was as follows: LE ~ I + BO + RB + LA + D, where: LE, standardized log life expectancy; I, insularity (binary); BO, standardized log body mass; RB, relative brain size; LA, standardized maximum latitude range and D, protein content diet (ordinal). Relative brain size was calculated as: BR − pBR, where: BR, standardized log brain mass and pBR, predicted brain mass from a second model that ran simultaneously: pBR~BO. Relative brain size has been shown to correlate with innovation rates in birds [43], and we therefore use it as a proxy for cognitive flexibility. Our implementation is similar to residual brain size in multiple regressions, but since both models are evaluated at each step of the sampling, information flows in both directions and measurement error is modelled correctly [44]. We included standard error around the mean for life expectancy, body mass and brain mass. We also included a phylogenetic variance–covariance matrix based on the phylogenetic distances calculated from Burgio *et al.* [42], using the L2-norm. For each variable with missing data, missing values were imputed using a multinormal distribution with mean and standard deviation based on the observed data, variance–covariance based on the phylogenetic signal and means further informed by the causal relationships outlined in figure 1. For life expectancy we had data for 244 species, but the models only converged for 217 species. Life expectancy for the remaining 143 parrot species was therefore imputed (see the electronic supplementary material, methods).

To test whether any correlation between relative brain size and longevity could be indirectly caused by developmental time, delayed juvenile periods and/or parental investment, we ran a second model (model 2) where developmental time (incubation period plus fledging period in model 2) and clutch size were included as additional covariates. Both variables were log transformed and standardized. Because data on AFR (a third measure of developmental time) were only available for 89 species and the available data were biased towards later AFR (see the electronic supplementary material, methods for more details), we did not attempt to impute this variable but tested

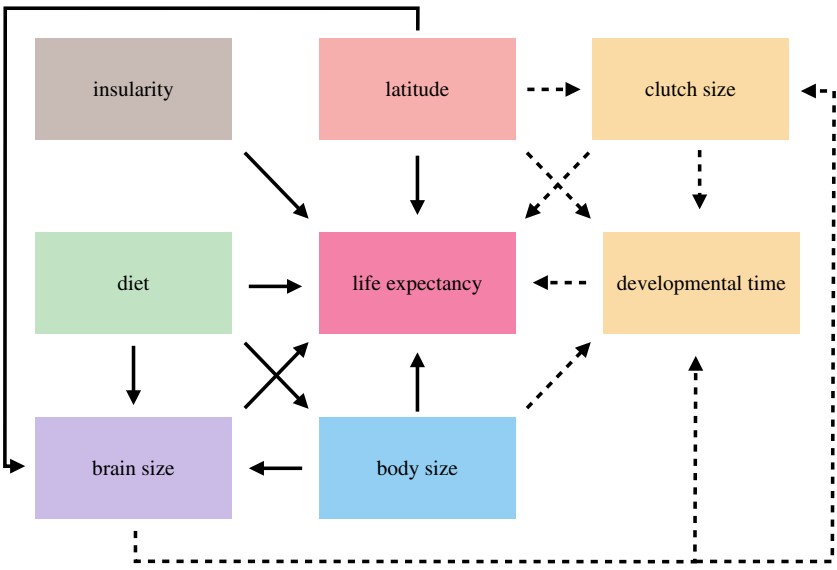

**Figure 1.** Directed acyclic graph of the potential causal pathways that could drive parrot life expectancy. Colours represent different covariate groups and are kept consistent throughout the manuscript. Solid lines represent assumed causal effects in all models (see §2c for model definitions). Dashed lines represent additional causal relationships in models 2 and 3. (Online version in colour.)

its effect in a third model (model 3) limited to cases where AFR was known.

To assess which hypothesis was best supported by the data, we compared the effect of relative brain size in the three models. If an increase in relative brain size directly causes an increase in life expectancy (*cognitive buffer hypothesis*), we would expect the coefficient of the brain size effect to be positive and similar in all three models. If an increase in relative brain size only causes an increase in developmental time (*expensive brain hypothesis*), we would expect the coefficient of the brain size effect to be positive in model 1 and zero in models 2 and 3. We would also expect an effect of developmental variables in models 2 and 3.

1, $\beta = 0.18$ in model 2 and $\beta = 0.16$ in model 3; overlap with zero less than 0.03 for all models; figures 3a and 4). Of the other life-history variables included, none appeared to have a large effect on life expectancy (figure 3d–h). In particular, model 2 showed no effect of developmental time ($\beta = 0.01$, overlap with zero greater than 0.22) or clutch size ($\beta = -0.08$, overlap with zero greater than 0.88) on longevity, and there was no clear effect of AFR on longevity in model 3 ($\beta = -0.11$, overlap with zero greater than 0.88). However, it should be noted that these models were designed to test the effect of relative brain size, so other parameter estimates should be interpreted with caution [45].

# 3. Results

Overall, we were able to estimate life expectancy for 217 species of 244 species for which we had data (for all other species, life expectancy was imputed in the final models). This included representatives of all eight major genera (i.e. those with at least 10 species) and over half of the extant parrot species. The shortest-lived genera were the small-bodied *Psittaculirostris* and *Charmosyna*, e.g. with a life expectancy of less than 2 years for *Psittaculirostris desmarestii*. The longest-lived genera were the large-bodied *Ara* and *Cacatua*, e.g. with a life expectancy of more than 35 years for *Ara macao* (full distribution of values across the phylogenetic tree is shown in figure 2). Similarly, there was large variability in other covariates, e.g. with brain size ranging from 1 to 22 grams, and AFR ranging from 7 months to 6 years. There was a strong phylogenetic signal in life expectancy (figure 2b); however, covariance was generally low between species that belonged to different genera (figure 3c).

Model 1 (without developmental time and parental investment) as well as models 2 and 3 (including these potential indirect paths) had similar estimates for the direct effect of relative brain size. As expected, body size was strongly and positively correlated with life expectancy (figure 3c for model 2; electronic supplementary material, results for models 1 and 3). Relative brain size also had a small, but consistently positive, effect on life expectancy ($\beta = 0.22$ in model

# 4. Discussion

Using an extensive database from captive parrots, our study showed a clear positive correlation between relative brain size and life expectancy in parrots. We further tested two hypotheses to explain this observed correlation between relative brain size and life expectancy: the *cognitive buffer hypothesis* [15] and the *expensive brain hypothesis* [18]. Our results best supported a direct relationship between larger brains and longer life expectancy, as predicted under the *cognitive buffer hypothesis*. It should be noted that this result is also consistent with the *delayed benefits hypothesis* [17]. We would, however, also expect a strong effect of diet on life expectancy, since this hypothesis argues that long lifespans allow species to invest more time in learning foraging skills which require larger brains and only pays off with an extended juvenile period. To fully explore this hypothesis we would need data on postfledging parental care and future studies could additionally try to use process-based approaches (where evolution is modelled explicitly), such as generative inference [46] or Bayesian ancestral state reconstruction [47] to disentangle the direction of causality. We found no evidence that the relationship between relative brain size and life expectancy was explained by the need for longer development times (here measured by incubation to fledging time and by age of first reproduction) or by increased parental

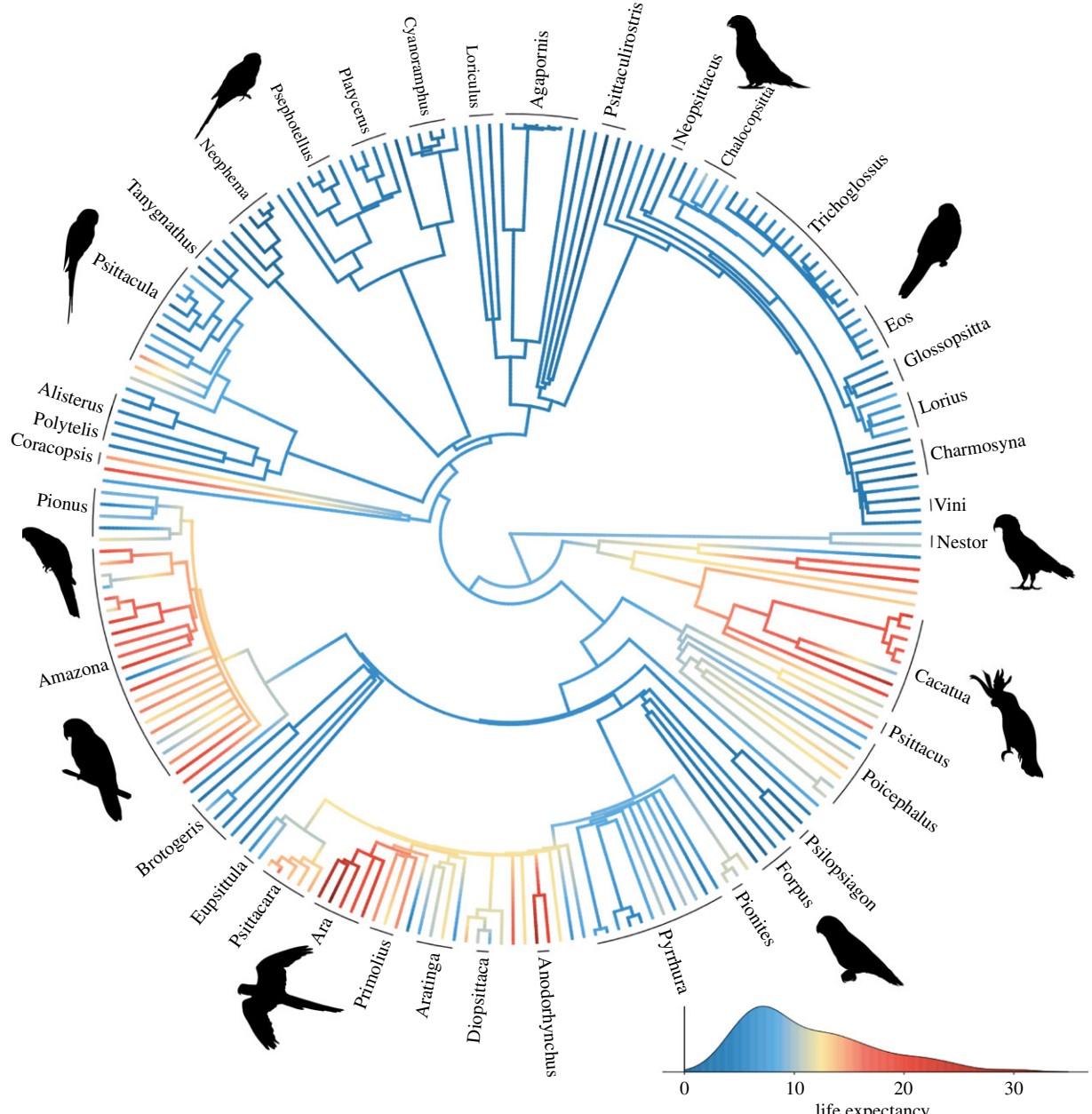

**Figure 2.** Phylogenetic tree of the 217 parrot species included in the study. Branches are coloured according to life expectancy (see density plot in bottom right), and phylogeny is based on Burgio *et al.* [42]. Genera are named if they contain at least two species. For a version with all species named, see the electronic supplementary material, figure S1. (Online version in colour.)

investment (here represented by clutch size), as predicted by the *expensive brain hypothesis*. Interestingly, our results differ from a previous study in parrots by Munshi-South & Wilkinson [19]. This study found that the protein content of diets and communal roosting best explained variation in maximum longevity. Data on sociality are largely lacking for parrots, so we did not test for an effect of sociality, but we found no effect of diet. However, Munshi-South & Wilkinson [19] did not consider brain size in their analysis. Because diet potentially determines whether and how quickly brains can grow [48], protein intake could still have an indirect effect on longevity via its potential link with brain size.

The lack of support for the *expensive brain hypothesis* is contrary to previous studies in primates [11,49], other mammals [29,50] and amphibians [10], all of which show a positive correlation between developmental time or AFR and life expectancy. However, it is in line with previous work examining the evolution of longevity in birds [14]. To

explain this discrepancy between birds and mammals, Isler *et al.* [16] suggested that bird species with allomaternal care (care provided for mother or offspring by either the father or helpers) can provide enough nutrition for relatively larger brained offspring without the need to prolong developmental periods or reduce clutch size to an extent that would lead to the coevolution of increased lifespans. All parrots have relatively large brain sizes compared to most other birds, and all parrot species exhibit biparental care. Almost all parrots are also cavity nesters. Cavity nests are less vulnerable to predation and often have extensive nest defence strategies and so can have relatively relaxed selective pressure on fledging times as compared to open-cup nesters [51]. Perhaps the combination of these factors provides enough flexibility to deal with heightened nutritional demands of rearing large-brained offspring without selection on developmental times. This does not, however, diminish the importance of cognitive development in parrots. The

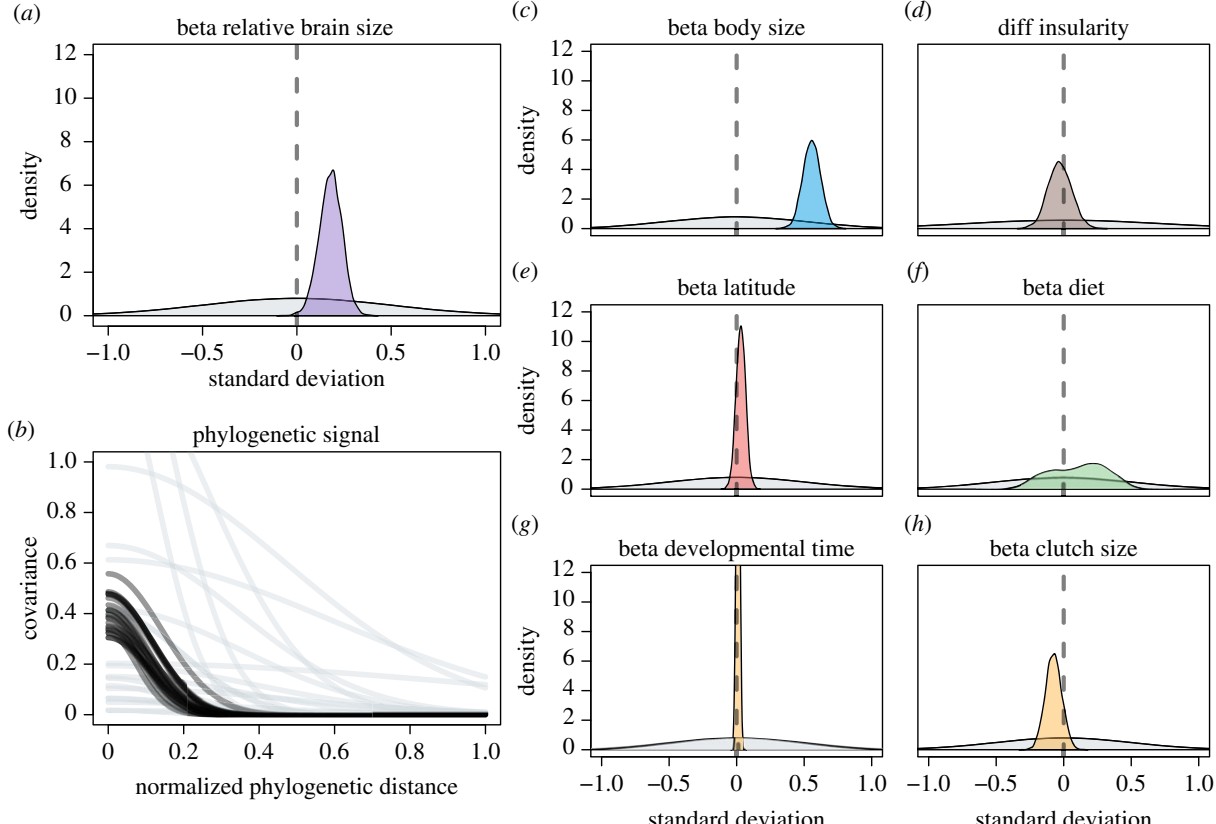

**Figure 3.** Parameter estimates for model 2. For results of models 1 and 3, see the electronic supplementary material, figures S2 and S4. Grey density plots and lines are the regularizing priors. Coloured areas are the posterior densities for the parameter estimates controlling the effect of the covariates on life expectancy. Black lines are 20 samples of the posterior for the phylogenetic covariance. For insularity, the difference between islandic and continental species is shown. (Online version in colour.)

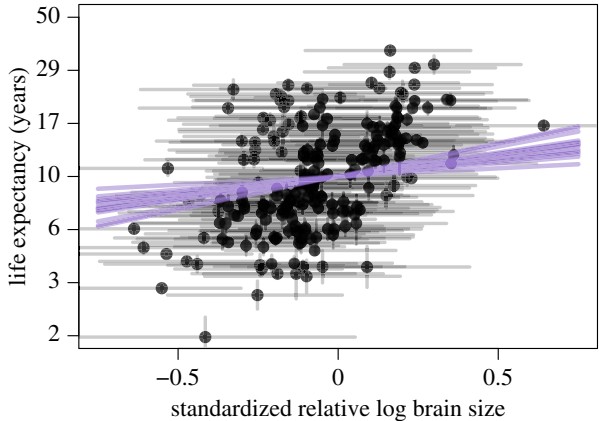

**Figure 4.** Standardized relative log brain size versus life expectancy for model 2. Black points represent 217 species where life expectancy was available, vertical black lines represent the s.e. for life expectancy, horizontal black lines represent the 89% percentile intervals for standardized relative log brain size. Purple lines represent 20 samples from the posterior for the slope (beta) of the effect of standardized relative log brain size on life expectancy. (Online version in colour.)

extended juvenile periods observed in many parrot species of up to 6 years may provide enhanced opportunities for social learning, as proposed for another large-brained bird taxon, the corvids [52]. This hypothesis remains to be tested in parrots.

To our knowledge, this is the first study of life expectancy and/or brain size that uses a bespoke Bayesian model to include the following: (i) uncertainty about variable estimates; (ii) imputation of missing values; (iii) a principled representation of relative brain size; and (iv) phylogenetic signal. We believe this method has some major advantages. Most notably, we could estimate both life expectancy and its uncertainty in each species. This allowed us to fully exploit the fact that we have a hundred-fold more data for some species, instead of relying on a single-point estimate of maximum longevity as in previous studies of longevity in parrots [19,27]. We also imputed life expectancy for species which have no data. This is likely to be important in most datasets to account for biased data collection, but it is especially important when using data from captivity, because zoos do not randomly pick species to be included in their population, but have a general bias towards larger and longer lived species [53]. Complete case analysis will introduce bias in this case [54], and we therefore chose to impute missing values. The use of DAGs and structural equation models is very similar to path analysis. The main advantage of our implementation is that it allows for a statistically robust definition of relative brain size and can handle uncertainty and missing data. Our model structure can be easily adapted to impute any continuous variable.

Our study also departs from most previous studies of longevity by using data from captivity on life expectancy [9,55–57]. This provided several important advantages. First, it provided a large sample size, both improving the estimation of life expectancy per species and allowing us to have a fuller representation of species. Second, captivity reduces external sources of mortality as much as possible (little

predation, starvation, etc.). However, captive data pose different challenges. First, as with data from the wild, birth and death dates can be missing (e.g. for individuals born in the wild or transferred from institutions that are not part of ZIMS). The BaSTA implementation that we used imputed these missing values, and we believe that our thorough cleaning procedure, coupled with the sheer magnitude of the dataset, means that any gaps, data entry errors or biases should have minimal effect on the life expectancies presented here. Second, there may be differences in causes of death in captivity and the wild, for example if some species are difficult to keep or prone to negative behavioural responses to captivity which is also true for some of the shortest-lived genera included in the study such as *Psittaculirostris* and *Charmosyna* which have been historically difficult to manage in captivity. We dealt with this by excluding potentially problematic species from the initial life expectancy estimations, and instead imputed values in the final model (see the electronic supplementary material, methods for details). We can still not be completely sure that the patterns observed in the data are all representative of the evolutionary processes that shaped them, but it is highly unlikely that the clear positive correlation between relative brain size and life expectancy is owing to captivity. It could even be expected that large-brained species live shorter in captivity, because of the higher metabolic rates required to keep the large brain supplied with glucose. This has been shown to be the case within species in captive guppies [12]. Since such an effect would be opposite to the one observed in our study, its presence would not change the conclusions drawn from our results.

## 5. Conclusion

Overall, our results are consistent with the *cognitive buffer hypothesis*, suggesting that relatively large brains may have buffered parrots against environmental variability and/or predation threats reducing sources of extrinsic mortality and allowing longer lifespans. This result is consistent with previous studies in other birds, suggesting that common processes may explain longevity in altricial birds. In addition to their longevity, parrots are famous for their complex cognition. It remains largely unknown what evolutionary processes have driven cognitive evolution in parrots, but given the results of our study, in addition to those of

Munshi-South & Wilkinson [19], future work should further investigate the potentially complex feedbacks between these two factors and sociality and diet. Unfortunately, longer lived species are also more likely to be threatened [27], showing the vulnerability of this order. Having life expectancy and other life-history variables for hundreds of species will hopefully aid in future conservation efforts for this globally threatened order.

Data accessibility. Data, code and materials are available from the Dryad Digital Repository: https://doi.org/10.5061/dryad.sbcc2fr7x [58] and the GitHub repository: https://github.com/simeonqs/Coevolution_of_relative_brain_size_and_life_expectancy_in_parrots.

Authors' contributions. S.Q.S.: conceptualization, data curation, formal analysis, investigation, methodology, project administration, visualization, writing—original draft, writing—review and editing; D.A.C.: conceptualization, funding acquisition, resources, supervision, writing—original draft, writing—review and editing; A.B.: methodology, writing—original draft, writing—review and editing; S.B.: methodology, writing—review and editing; A.I.: data curation, resources, writing—review and editing; J.S.: data curation, methodology, writing—review and editing; T.F.W.: data curation, writing—original draft; A.M.Y.: conceptualization, data curation, writing—review and editing; M.B.M.: conceptualization, formal analysis, funding acquisition, resources, supervision, writing—original draft, writing—review and editing; L.A.: conceptualization, formal analysis, funding acquisition, resources, supervision, writing—original draft, writing—review and editing.

All authors gave final approval for publication and agreed to be held accountable for the work performed therein.

Competing interests. The authors have no competing interests with this study.

Funding. Open access funding provided by the Max Planck Society.

This project was possible thanks to the financial support of the sponsor members of the Species360 Conservation Science Alliance (Copenhagen Zoo, the World Association of Zoos and Aquariums and the Wildlife Reserves of Singapore), the Interdisciplinary Centre on Population Dynamics and the Biology Department at the University of Southern Denmark. We would like to thank Species360 for granting access to the data under permission number no. 86892. This work was supported by the Max Planck Society. L.A. was funded by a Max Planck Independent Group Leader Fellowship. S.Q.S. received additional funding from the International Max Planck Research School for Organismal Biology.

Acknowledgements. We would like to thank Dr Dieter Lukas, Dr Cody Ross, Prof. Fernando Colchero, Prof. Richard McElreath and Dr Rita da Silva for their advice on the analysis. We would further like to thank the more than 1300 Species360 members for registering their animals in the ZIMS database. We thank Prof. Alejandro Salinas, Geddes Hislop and Johann C. Carstens for contributing data on diet.

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
