## [Peer Review File · Proceedings of the Royal Society B: Biological Sciences]

Review History

RSPB-2021-2397.R0 (Original submission)

Review form: Reviewer 1

Recommendation

Accept with minor revision (please list in comments)

Scientific importance: Is the manuscript an original and important contribution to its field?

Excellent

General interest: Is the paper of sufficient general interest?

Excellent

Quality of the paper: Is the overall quality of the paper suitable?

Good

Is the length of the paper justified?

Yes

Should the paper be seen by a specialist statistical reviewer?

Yes

Do you have any concerns about statistical analyses in this paper? If so, please specify them explicitly in your report.

No

It is a condition of publication that authors make their supporting data, code and materials available - either as supplementary material or hosted in an external repository. Please rate, if applicable, the supporting data on the following criteria.

Is it accessible?

Yes

Is it clear?

Yes

Is it adequate?

Yes

Do you have any ethical concerns with this paper?

No

Comments to the Author

Dear Editor,

In this study Smeele et al. relate life expectancy of parrots in captivity to their relative brain size and find that larger-brained parrots indeed live longer. They use a new statistical method to do so and it looks exciting. However, even though I agree with the logic of what has been done I cannot judge whether the methods are adequate as this is beyond my expertise. I trust other referees will be better qualified to scrutinize everything Bayesian in this manuscript. I enjoyed reading the manuscript and am sure it will be interesting to many. Of course, I did find some aspects that could be considered in a revision. One main one and two smaller ones:

My main point of criticism is that the manuscript is one-sided as the predictions for the relationship between brain size and longevity only include positive associations. There would be reasons to also predict the other way around. This is based on some reasoning about how intrinsic and extrinsic causes of death impact the evolution of ageing. Details for instance in (Chen & Maklakov 2012). Applied to this manuscript, life expectancy in captivity reflects how long animals live 'intrinsically' - being clever or not should not make a big difference in a cage as also less clever individuals will be fed and cared for adequately. All extrinsic causes of death are removed (predation, starvation,...). Studies in fish showed that while in a semi-wild setting large-brained animals live longer (Kotrschal et al. 2015), large-brained animals actually lived shorter in captivity (Kotrschal, Corral-Lopez & Kolm 2019). The authors argued that when all opportunities to use the extra computing capacity of a larger brain to survive tricky situations is removed in captivity, the costs of a large brain become evident. Brain size may be traded off against investments into defenses against ageing. This is an alternative prediction for the relationship between brain size and longevity in captivity that actually suggests an inverse relationship. I leave it up to the authors whether they want to change the introduction accordingly or bring it up in the discussion, but I think that this apparent discrepancy between fishes and birds should be discussed. Ideally with some links to the evolution of ageing literature. Line 28-30: Is that a good justification to do the study - that it has not been done in parrots? It has not been done in most animal groups. I suggest reshuffling the first sentences and start with longevity and not with parrots. Like it is done in the introduction.

Line 56: The link between brain size and cognitive flexibility/ability in general should be explained and backed up by references. What is the evidence that brain size is a proxy for any cognitive aspect in parrots?

Methods: some parrot brains sizes were based on actual brain weights, others on endocasts. Did the authors check whether the method of brain size estimation impact the results?

Chen, H.-y. & Maklakov, A. A. 2012. Longer life span evolves under high rates of condition-dependent mortality. *Current Biology*, 22, 2140-2143.

Kotrschal, A., Buechel, S. D., Zala, S. M., Corral-Lopez, A., Penn, D. J. & Kolm, N. 2015. Brain size affects female but not male survival under predation threat. *Ecology Letters*, 18, 646-652.

Kotrschal, A., Corral-Lopez, A. & Kolm, N. 2019. Large brains, short life: selection on brain size impacts intrinsic lifespan. *Biology Letters*, 15, 20190137.

Review form: Reviewer 2

Recommendation

Major revision is needed (please make suggestions in comments)

Scientific importance: Is the manuscript an original and important contribution to its field?

Excellent

General interest: Is the paper of sufficient general interest?

Acceptable

Quality of the paper: Is the overall quality of the paper suitable?

Good

Is the length of the paper justified?

Yes

Should the paper be seen by a specialist statistical reviewer?

Yes

Do you have any concerns about statistical analyses in this paper? If so, please specify them explicitly in your report.

Yes

It is a condition of publication that authors make their supporting data, code and materials available - either as supplementary material or hosted in an external repository. Please rate, if applicable, the supporting data on the following criteria.

Is it accessible?

Yes

Is it clear?

Yes

Is it adequate?

Yes

Do you have any ethical concerns with this paper?

No

Comments to the Author

Please see attached. (See Appendix A)

Decision letter (RSPB-2021-2397.R0)

07-Dec-2021

Dear Mr Smeele:

Your manuscript has now been peer reviewed and the reviews have been assessed by an Associate Editor. The reviewers' comments (not including confidential comments to the Editor) and the comments from the Associate Editor are included at the end of this email for your reference. As you will see, the reviewers and the Editors have raised some concerns with your manuscript and we would like to invite you to revise your manuscript to address them.

Research ethics:

Use of animals and field studies:

It is a condition of publication that you make available the data and research materials supporting the results in the article. Please see our Data Sharing Policies (<https://royalsociety.org/journals/authors/author-guidelines/#data>). Datasets should be deposited in an appropriate publicly available repository and details of the associated accession number, link or DOI to the datasets must be included in the Data Accessibility section of the

article (<https://royalsociety.org/journals/ethics-policies/data-sharing-mining/>). Reference(s) to datasets should also be included in the reference list of the article with DOIs (where available).

Please submit a copy of your revised paper within three weeks. If we do not hear from you within this time your manuscript will be rejected. If you are unable to meet this deadline please let us know as soon as possible, as we may be able to grant a short extension.

Best wishes,
Dr Locke Rowe
mailto: proceedingsb@royalsociety.org

Associate Editor
Comments to Author:

Thank you for your patience. We have received comments from two reviewers that, while being globally supportive, have requested further clarification on the hypotheses and statistical tests employed. Please review these comments. Pending satisfactorily addressing these issues, we will look forward to your resubmission.

Reviewer(s)' Comments to Author:
Referee: 1

Comments to the Author(s)
Dear Editor,

In this study Smeele at al. relate life expectancy of parrots in captivity to their relative brain size and find that larger-brained parrots indeed live longer. They use a new statistical method to do so and it looks exciting. However, even though I agree with the logic of what has been done I

cannot judge whether the methods are adequate as this is beyond my expertise. I trust other referees will be better qualified to scrutinize everything Bayesian in this manuscript. I enjoyed reading the manuscript and am sure it will be interesting to many. Of course, I did find some aspects that could be considered in a revision. One main one and two smaller ones:

My main point of criticism is that the manuscript is one-sided as the predictions for the relationship between brain size and longevity only include positive associations. There would be reasons to also predict the other way around. This is based on some reasoning about how intrinsic and extrinsic causes of death impact the evolution of ageing. Details for instance in (Chen & Maklakov 2012). Applied to this manuscript, life expectancy in captivity reflects how long animals live 'intrinsically' – being clever or not should not make a big difference in a cage as also less clever individuals will be fed and cared for adequately. All extrinsic causes of death are removed (predation, starvation,...). Studies in fish showed that while in a semi-wild setting large-brained animals live longer (Kotrschal et al. 2015), large-brained animals actually lived shorter in captivity (Kotrschal, Corral-Lopez & Kolm 2019). The authors argued that when all opportunities to use the extra computing capacity of a larger brain to survive tricky situations is removed in captivity, the costs of a large brain become evident. Brain size may be traded off against investments into defenses against ageing. This is an alternative prediction for the relationship between brain size and longevity in captivity that actually suggests an inverse relationship. I leave it up to the authors whether they want to change the introduction accordingly or bring it up in the discussion, but I think that this apparent discrepancy between fishes and birds should be discussed. Ideally with some links to the evolution of ageing literature.

Line 28-30: Is that a good justification to do the study – that it has not been done in parrots? It has not been done in most animal groups. I suggest reshuffling the first sentences and start with longevity and not with parrots. Like it is done in the introduction.

Line 56: The link between brain size and cognitive flexibility/ability in general should be explained and backed up by references. What is the evidence that brain size is a proxy for any cognitive aspect in parrots?

Methods: some parrot brains sizes were based on actual brain weights, others on endocasts. Did the authors check whether the method of brain size estimation impact the results?

Chen, H.-y. & Maklakov, A. A. 2012. Longer life span evolves under high rates of condition-dependent mortality. *Current Biology*, 22, 2140-2143.

Kotrschal, A., Buechel, S. D., Zala, S. M., Corral-Lopez, A., Penn, D. J. & Kolm, N. 2015. Brain size affects female but not male survival under predation threat. *Ecology Letters*, 18, 646-652.

Kotrschal, A., Corral-Lopez, A. & Kolm, N. 2019. Large brains, short life: selection on brain size impacts intrinsic lifespan. *Biology Letters*, 15, 20190137.

Referee: 2

Comments to the Author(s)

Please see attached.

Author's Response to Decision Letter for (RSPB-2021-2397.R0)

See Appendix B.

RSPB-2021-2397.R0 (Original submission)

Review form: Reviewer 1

Recommendation

Accept as is

Scientific importance: Is the manuscript an original and important contribution to its field?

Good

General interest: Is the paper of sufficient general interest?

Good

Quality of the paper: Is the overall quality of the paper suitable?

Good

Is the length of the paper justified?

Yes

Should the paper be seen by a specialist statistical reviewer?

No

Do you have any concerns about statistical analyses in this paper? If so, please specify them explicitly in your report.

No

It is a condition of publication that authors make their supporting data, code and materials available - either as supplementary material or hosted in an external repository. Please rate, if applicable, the supporting data on the following criteria.

Is it accessible?

Yes

Is it clear?

Yes

Is it adequate?

Yes

Do you have any ethical concerns with this paper?

No

Comments to the Author

I am fully content with the author's responses to my suggestions. Well done on a really interesting and accessible paper.

Review form: Reviewer 2

Recommendation

Accept with minor revision (please list in comments)

Scientific importance: Is the manuscript an original and important contribution to its field?

Good

General interest: Is the paper of sufficient general interest?

Good

Quality of the paper: Is the overall quality of the paper suitable?

Excellent

Is the length of the paper justified?

Yes

Should the paper be seen by a specialist statistical reviewer?

No

Do you have any concerns about statistical analyses in this paper? If so, please specify them explicitly in your report.

No

It is a condition of publication that authors make their supporting data, code and materials available - either as supplementary material or hosted in an external repository. Please rate, if applicable, the supporting data on the following criteria.

Is it accessible?

Yes

Is it clear?

Yes

Is it adequate?

Yes

Do you have any ethical concerns with this paper?

No

Comments to the Author

Please see attached.

Decision letter (RSPB-2021-2397.R1)

17-Feb-2022

Dear Mr Smeele

I am pleased to inform you that your manuscript RSPB-2021-2397.R1 entitled "Coevolution of relative brain size and life expectancy in parrots" has been accepted for publication in Proceedings B.

The referee(s) have recommended publication, but also suggest some minor revisions to your manuscript. Therefore, I invite you to respond to the referee(s)' comments and revise your manuscript. Because the schedule for publication is very tight, it is a condition of publication that you submit the revised version of your manuscript within 7 days. If you do not think you will be able to meet this date please let us know.

[http://datadryad.org/submit?journalID=RSPB&manu=\(Document not available\)](http://datadryad.org/submit?journalID=RSPB&manu=(Document not available)) which will take you to your unique entry in the Dryad repository. If you have already submitted your data to dryad you can make any necessary revisions to your dataset by following the above link.

Please see <https://royalsociety.org/journals/ethics-policies/data-sharing-mining/> for more details.

Sincerely,
Dr Locke Rowe
Editor, Proceedings B
<mailto:proceedingsb@royalsociety.org>

Associate Editor:

Comments to Author:

Thank you for your submission. Your article has received two positive reviews with reviewer 2 offering minor revisions. Please look over these comments and address them accordingly.

Reviewer(s)' Comments to Author:

Referee: 1

Comments to the Author(s)

I am fully content with the author's responses to my suggestions. Well done on a really interesting and accessible paper.

Referee: 2

Comments to the Author(s)

Please see attached.

Author's Response to Decision Letter for (RSPB-2021-2397.R1)

See Appendix C.

Decision letter (RSPB-2021-2397.R2)

21-Feb-2022

Dear Mr Smeele

I am pleased to inform you that your manuscript entitled "Coevolution of relative brain size and life expectancy in parrots" has been accepted for publication in Proceedings B.

Your article has been estimated as being 8 pages long. Our Production Office will be able to confirm the exact length at proof stage.

Data Accessibility section

Open Access

Paper charges

Sincerely,

Proceedings B

Appendix A

Comments to the Author

Review of “Coevolution of brain size and longevity in parrots” (RSPB-2021-2397)

This is an interesting manuscript that expands our understanding of an important evolutionary phenomenon – the apparent coevolution of relative brain size and longevity observed across and within many animal groups. The goal of this manuscript is to demonstrate whether this association is present in parrots and, if so, whether this relationship reflects large brains buffering of ecological challenges (‘cognitive buffering’) or longer lifespans compensating for extended development and/or parental investment (i.e., ‘expensive brain’). To accomplish this, they gather an impressively large life expectancy data set and test whether life expectancy is predicted by body size, relative brain size, environmental variables (latitude, diet, insularity), or developmental variables (developmental time, clutch size). The manuscript is well-written and uses a novel data set and approach. I recommend major revisions to this manuscript, due to the need for a more detailed description of the hypotheses tested and statistical framework employed.

My comments are organized by section below:

Title

- The title should more directly and accurately describe the results (as *relative* brain size and *life expectancy* are the actual variables being explicitly tested). For example: ‘Cognitive buffering explains the coevolution of relative brain size and life expectancy in parrots’

Introduction

- Lines 100-101: This is repeating a point made in Lines 96-97.
- Line 90: I do not believe DeCasien et al. included residuals in regression models, but only for visualization (as in Figure 4 of this manuscript)
- The delayed benefits hypothesis has been applied to the evolution of non-hominid species (e.g., birds: Sol et al. 2016) and is not irrelevant to the analysis presented. Accordingly, this hypothesis should not be dismissed in the Introduction and the results should be discussed in terms of this hypothesis (in addition to cognitive buffering).

Methods

- It is not explicitly stated here (or anywhere else in the manuscript) how the different models tested would support/contradict the different hypotheses under consideration. This is critical for the flow of the paper and interpretation of the results presented. For example: “If relative brain size is a significant positive predictor in Model 1 but not in Models 2/3, this would suggest that the contribution of relative brain size to life expectancy is driven by developmental variables.”
 - Along these lines, correlations between all predictors should be analyzed and presented to: 1) provide a fuller description of the comparative dataset gathered here; and 2) inform the readers as to whether collinearity between predictors may impact the model results (e.g., VIFs).
- The other main issue I see with the manuscript is that there is far from enough description of the statistical modelling approach employed here. Comparative biologists who employ typical

phylogenetic statistical models (e.g., PGLS, Bayesian phylogenetic mixed models in MCMCglmm, etc.) will not be familiar with the approach used here. Examples of details to be clarified include:

- What is L2-norm? What are backdoor criteria?
- How is this approach similar to and different from more popular methods?
- How exactly does the DAG inform the regression analyses?
 - In line with this, focusing on the DAG (here and in the Introduction) and the use of “direct” and “indirect” effects makes it seem like a path analysis is going to be used, but this is not the case. Please elaborate.
- Why not include body and brain size in the models (rather than body and relative brain size, with the latter estimated using a simultaneously run model)? These measures are conceptually different (i.e., brain size relative to body size vs. relative brain size relative to body size) and should be justified.
- How were the priors described in the supplement chosen?
- Are the results similar if a more familiar approach (e.g., PGLS on species mean values) is employed?
- More description is needed for the various predictor variables (e.g., how were protein diet levels calculated?)
- Why isn't AFR include in the DAG?
- Figure 1 includes references to models 1,2,3 but this figure is referred to before these models are described.
- I suggest numbering the models (e.g., following model descriptions or equations with “(Model 1)”) in the Methods (and also in the Results) to better guide readers.
- Line 179: Here it is stated that 360 species were included in the models but earlier it is stated that 244 species had cleaned life expectancy data available and that these data converged for only 217 species? Please clarify.

Results

- Lines 224-225: How exactly were these models “designed to test the effect of relative brain size”? Shouldn't any potential significant effects of non-brain variables be detectable in these models (if they exist)?
- These results only include coefficient estimates but not significance values (e.g., the proportion of values that cross 0) – such values should be estimated and added to indicate confidence in effect size estimates.

Discussion

- I appreciate the thoughtful discussion on the (lack of) diet associations.
-

Figures

- Figure 4: Why were only 20 samples from the posterior depicted (rather than mean and interval)? The latter may be more appropriate.

Supplementary Materials

- Model 3 is missing from the Supplement.

I hope that the Authors find my comments helpful in revising their manuscript.

Appendix B

Associate Editor

Comments to Author:

Thank you for your patience. We have received comments from two reviewers that, while being globally supportive, have requested further clarification on the hypotheses and statistical tests employed. Please review these comments. Pending satisfactorily addressing these issues, we will look forward to your resubmission.

Please submit a copy of your revised paper within three weeks. If we do not hear from you within this time your manuscript will be rejected. If you are unable to meet this deadline please let us know as soon as possible, as we may be able to grant a short extension.

Best wishes,
Dr Locke Rowe

Dear Dr Locke Rowe,

Thank you very much for considering our manuscript for publication. We are happy to receive so much constructive feedback and have done our best to incorporate as many of their comments as possible. We address their comments line by line below.

We believe that the current version is very much improved, and we are looking forward to hearing from you again.

Yours sincerely,
Simeon Smeele (on behalf of all authors)

Reviewer(s)' Comments to Author: Referee 1

Dear Editor,

In this study Smeele et al. relate life expectancy of parrots in captivity to their relative brain size and find that larger-brained parrots indeed live longer. They use a new statistical method to do so and it looks exciting. However, even though I agree with the logic of what has been done I cannot judge whether the methods are adequate as this is beyond my expertise. I trust other referees will be better qualified to scrutinize everything Bayesian in this manuscript. I enjoyed reading the manuscript and am sure it will be interesting to many.

Thank you for your positive feedback on the manuscript.

Of course, I did find some aspects that could be considered in a revision. One main one and two smaller ones:

My main point of criticism is that the manuscript is one-sided as the predictions for the relationship between brain size and longevity only include positive associations. There would be reasons to also predict the other way around. This is based on some reasoning about how intrinsic and extrinsic causes of death impact the evolution of ageing. Details for instance in (Chen & Maklakov 2012).

Applied to this manuscript, life expectancy in captivity reflects how long animals live 'intrinsically' – being clever or not should not make a big difference in a cage as also less clever individuals will be fed and cared for adequately. All extrinsic causes of death are removed (predation, starvation,...). Studies in fish showed that while in a semi-wild setting large-brained animals live longer (Kotrschal et al. 2015), large-brained animals actually lived shorter in captivity (Kotrschal, Corral-Lopez & Kolm 2019). The authors argued that when all opportunities to use the extra computing capacity of a larger brain to survive tricky situations is removed in captivity, the costs of a large brain become evident. Brain size may be traded off against investments into defenses against ageing. This is an alternative prediction for the relationship between brain size and longevity in captivity that actually suggests an inverse relationship. I leave it up to the authors whether they want to change the introduction accordingly or bring it up in the discussion, but I think that this apparent discrepancy between fishes and birds should be discussed. Ideally with some links to the evolution of ageing literature.

Thank you very much for this input; the possibility of a negative relationship between brain size and longevity is a valid point that we had not considered in our manuscript. Overall, we think that our focus on an expectation of a positive correlation between brain size and longevity is justified, given the weight of previous evidence, particularly in birds. However, it is interesting to consider when we would expect the opposite, and how within-species studies inform this. While there isn't room to fully explore this in our manuscript, we have revised the first paragraph of our introduction to acknowledge this possibility, and included the suggested reference:

L57 (line numbers refer to Main Document): *“Perhaps of most recent interest, brain size has been correlated with longevity across diverse taxa ranging from amphibians (10) to primates (11). While some studies have proposed a negative relationship between brain size and longevity, suggesting a trade-off between the energetic costs into larger brains and investments in defences against ageing (e.g., (12)), the large majority of studies have suggested a positive effect of larger brain sizes on longevity (10,11,13–17).”*

In addition, you raise the interesting possibility that a captive effect may exist that would lead to larger brained individuals living shorter in captivity once extrinsic sources of mortality are removed. This a valid point, and we have added it to our paragraph on possible captive effects in the discussion. However, while it might change the strength our of results, we believe that this would not change the direction of our conclusions, as such an effect show rather weaken the positive correlation between brain size and life expectancy.

At L334, it now reads: *“It could even be expected that large brained species live shorter in captivity, because of the higher metabolic rates required to keep the large brain supplied with glucose. This has been shown to be the case within species in captive guppies (12). Since such an effect would be opposite to the one observed in our study, its presence would not change the conclusions drawn from our results.”*

Line 28-30: Is that a good justification to do the study – that it has not been done in parrots? It has not been done in most animal groups. I suggest reshuffling the first sentences and start with longevity and not with parrots. Like it is done in the introduction.

Thank you for the suggestion. We reshuffled the first sentences; it now reads:

L32: *“Previous studies have demonstrated a correlation between longevity and brain size in a variety of taxa. Little research has been devoted to understanding this link in parrots; yet parrots are well-known for both their exceptionally long lives and cognitive complexity.”*

Line 56: The link between brain size and cognitive flexibility/ability in general should be explained and backed up by references. What is the evidence that brain size is a proxy for any cognitive aspect in parrots?

This is a very valid point. We have added a reference on L203 showing that innovation rate (the most important measure of cognitive flexibility for our paper) is correlated with relative brain size across birds: Overington et al. (2009). Technical innovations drive the relationship between innovativeness and residual brain size in birds. *Anim Behav*.

In addition, we now explicitly mention this evidence in our methods. L203 now reads:

“Relative brain size has been shown to correlate with innovation rates in birds (42) and we therefore use it as a proxy for cognitive flexibility.”

Methods: some parrot brains sizes were based on actual brain weights, others on endocasts. Did the authors check whether the method of brain size estimation impact the results?

This is a good point. We use both total brain weight and endocasts. For the latter we calculate the corresponding brain weight. For studies in birds, Iwaniuk et al. 2020 showed that this should not impact results. We have included a short statement and the reference in the Supplemental Methods (page 2).

Reviewer(s)' Comments to Author: Referee 2

This is an interesting manuscript that expands our understanding of an important evolutionary phenomenon – the apparent coevolution of relative brain size and longevity observed across and within many animal groups. The goal of this manuscript is to demonstrate whether this association is present in parrots and, if so, whether this relationship reflects large brains buffering of ecological challenges (‘cognitive buffering’) or longer lifespans compensating for extended development and/or parental investment (i.e., ‘expensive brain’). To accomplish this, they gather an impressively large life expectancy data set and test whether life expectancy is predicted by body size, relative brain size, environmental variables (latitude, diet, insularity), or developmental variables (developmental time, clutch size). The manuscript is well-written and uses a novel data set and approach. I recommend major revisions to this manuscript, due to the need for a more detailed description of the hypotheses tested and statistical framework employed.

Thank you very much for the extensive comments. We have addressed all comments and think the result is much easier to understand for the broader scientific community.

My comments are organized by section below: Title

- The title should more directly and accurately describe the results (as *relative* brain size and *life expectancy* are the actual variables being explicitly tested). For example: ‘Cognitive buffering explains the coevolution of relative brain size and life expectancy in parrots’

We have updated the title. We refrained from claiming only support for the cognitive buffer hypothesis in the title, since the delayed benefit hypothesis could potentially also explain our data.

Introduction

- Lines 100-101: This is repeating a point made in Lines 96-97.

Thank you very much. It's great to be able to also reduce the length of the manuscript and get rid of redundancy. We removed lines 100-101 (line numbering from the old version).

- Line 90: I do not believe DeCasien et al. included residuals in regression models, but only for visualization (as in Figure 4 of this manuscript)

It is true that DeCasien does not include residuals directly, however, their binary variable is computed from residuals. As this might be confusing, we replaced the reference by another example (González-Lagos et al. 2010). Thank you for pointing this out.

- The delayed benefits hypothesis has been applied to the evolution of non-hominid species (e.g., birds: Sol et al. 2016) and is not irrelevant to the analysis presented. Accordingly, this hypothesis should not be dismissed in the Introduction and the results should be discussed in terms of this hypothesis (in addition to cognitive buffering).

We have rewritten part of the introduction (L71-77, line numbers refer to Main Document) and discussion (265-275) to better explain why we do not believe strong claims can be made about this hypothesis, while including our results on diet to explain why we think it unlikely that this hypothesis could explain the strong correlation between relative brain size and life expectancy in our data. We would need data on skill acquisition, post-fledging parental care and a process-based model to fully explore the predictions made by this hypothesis.

Methods

- It is not explicitly stated here (or anywhere else in the manuscript) how the different models tested would support/contradict the different hypotheses under consideration. This is critical for the flow of the paper and interpretation of the results presented. For example: "If relative brain size is a significant positive predictor in Model 1 but not in Models 2/3, this would suggest that the contribution of relative brain size to life expectancy is driven by developmental variables."

Thank you for pointing this out. We added a paragraph that explicitly mentions our predictions to the end of the methods section (L225-230).

- Along these lines, correlations between all predictors should be analyzed and presented to: 1) provide a fuller description of the comparative dataset gathered here; and 2) inform the readers as to whether collinearity between predictors may impact the model results (e.g., VIFs).

Collinearity between predictor variables does not affect estimates in a well specified Bayesian model (McElreath, Rethinking, 2nd Edition, Chapter 6.1). It could influence sampling efficiency. However, we ran four parallel chains and monitored the Rhat values, which would have highlighted any sampling issues. Since most variables will be correlated to some extent to body size, we do not believe it informative to include correlations between all predictors, as this could easily lead readers to find spurious correlations.

- The other main issue I see with the manuscript is that there is far from enough description of the statistical modelling approach employed here. Comparative biologists who employ typical phylogenetic statistical models (e.g., PGLS, Bayesian phylogenetic mixed models in MCMCglmm, etc.) will not be familiar with the approach used here. Examples of details to be clarified include:

- What is L2-norm? What are backdoor criteria?

Thank you for the question. We did not sufficiently explain the L2-norm and added additional explanation to the introduction (L191-192). We do, however, not have enough space to explain the back-door criterion. This is a well-established statistical approach, but it requires multiple paragraphs to explain in enough detail to avoid confusion. We included a reference (Pearl et al. 2018) that goes in depth with this method.

- How is this approach similar to and different from more popular methods? How exactly does the DAG inform the regression analyses?
 - In line with this, focusing on the DAG (here and in the Introduction) and the use of “direct” and “indirect” effects makes it seem like a path analysis is going to be used, but this is not the case. Please elaborate.

Using a DAG and estimating direct effects is essentially a path analysis. We chose to use the term *structural equation model* instead to not confuse it with standardized methods. We have added an explicit explanation to justify the choice of method (L312-315).

- Why not include body and brain size in the models (rather than body and relative brain size, with the latter estimated using a simultaneously run model)? These measures are conceptually different (i.e., brain size relative to body size vs. relative brain size relative to body size) and should be justified.

Relative brain size is the appropriate measure to use, since we are testing the effect of having larger brains than required for a certain body mass. If relative brain size was the response variable, this could be achieved by including whole brain size together with body mass as predictor, but this is not the case if relative brain size is a predictor variable. We do include body mass as well, since body mass has an effect on life expectancy independent of the allometric relation with brain size (e.g., predator protection).

- How were the priors described in the supplement chosen?

We included the priors for the three main models under a separate heading in the Supplemental Methods (page 8). In general priors were chosen such that they allowed all reasonable values, while mildly regularizing towards a conservative effect.

- Are the results similar if a more familiar approach (e.g., PGLS on species mean values) is employed?

Since PGLS cannot be run with missing data or relative brain size, we would not expect similar results. Removing species with missing data leads to bias unless data is missing

‘completely at random’, which is not the case for our data. In addition, using residual values as data has been shown to lead to incorrect inference. For these reasons, we chose to use a Bayesian approach that can handle missing values and data uncertainty without compromising inference.

More description is needed for the various predictor variables (e.g., how were protein diet levels calculated?)

We tried to balance brevity and completeness in the main manuscript. We have added a detailed description of all the decisions to calculate each variable to the Supplemental Methods (see page 3). We include a small description for each variable in the main text, to give an idea of what each variable represents.

Why isn't AFR include in the DAG?

AFR is a developmental variable and is represented by this box in the DAG. We ran two separate models (2 and 3), since we did not have enough high-quality data on AFR to run a model on the full set of species with imputation of missing values. We choose not to include AFR as a separate box to increase readability of the graph. We realize this is not clear in the methods section and have added more clarification to the text (L220-223).

- Figure 1 includes references to models 1,2,3 but this figure is referred to before these models are described.

Thank you for pointing this out. We have added a reference to the appropriate methods section to the figure. Since we can't talk about the models without the figure, we need to refer to a later section rather than move the figure.

- I suggest numbering the models (e.g., following model descriptions or equations with "(Model 1)") in the Methods (and also in the Results) to better guide readers.

We have added model names more explicitly throughout the text.

- Line 179: Here it is stated that 360 species were included in the models but earlier it is stated that 244 species had cleaned life expectancy data available and that these data converged for only 217 species? Please clarify.

We recognize that these numbers are confusing. We clarified it in the text (L213-214).

Results

- Lines 224-225: How exactly were these models “designed to test the effect of relative brain size”? Shouldn't any potential significant effects of non-brain variables be detectable in these models (if they exist)?

Not necessarily. When deconfounding the path between X and Y the aim is to achieve an unbiased estimate of the effect of X on Y, but the effect of any Z on Y can still be confounded. Our paper aimed to test the direct and indirect effect of relative brain size on life expectancy and we used the back-door criterion to choose the appropriate set of deconfounders for our models. We included an additional reference with a good description of our approach (Pearl et al. 2018).

- These results only include coefficient estimates but not significance values (e.g., the proportion of values that cross 0) – such values should be estimated and added to indicate confidence in effect size estimates.

We initially avoided including these values as they can easily be confused with p-values. We have now added them to the results (L250-254).

Discussion

- I appreciate the thoughtful discussion on the (lack of) diet associations.

Thank you for the kind words. It's nice to have some positive feedback.

Figures

- Figure 4: Why were only 20 samples from the posterior depicted (rather than mean and interval)? The latter may be more appropriate.

We choose to display 20 samples (or full distributions for single parameters) to give a more balanced picture of the posterior distribution. By displaying only a single interval, a lot of emphasis is put on an arbitrarily chosen value. Bayesian posterior distributions do not need to be normally distributed and the shape of the posterior is actually informative in itself. For outcomes that depend on two or more variables (such as the lines in Figure 4), this is even more distinct, as it can be hard to distinguish uncertainty on the intercept from uncertainty from the slope when only displaying an interval. Figure 4 shows more clearly that there is high certainty about the intercept, but some uncertainty about the slope.

Supplementary Materials

- Model 3 is missing from the Supplement.

Thank you for pointing this out. The model was only mentioned briefly, so we added headers to make each model easier to find.

I hope that the Authors find my comments helpful in revising their manuscript.

Appendix C

Associate Editor

Comments to Author:

I am pleased to inform you that your manuscript RSPB-2021-2397.R1 entitled "Coevolution of relative brain size and life expectancy in parrots" has been accepted for publication in Proceedings B.

The referee(s) have recommended publication, but also suggest some minor revisions to your manuscript. Therefore, I invite you to respond to the referee(s)' comments and revise your manuscript. Because the schedule for publication is very tight, it is a condition of publication that you submit the revised version of your manuscript within 7 days. If you do not think you will be able to meet this date please let us know.

...

Sincerely,
Dr Locke Rowe
Editor, Proceedings B

Dear Dr Locke Rowe,

Thank you very much accepting our manuscript with minor revisions. We appreciate the additional comments and have addressed them in the main text and responded below.

Yours sincerely,
Simeon Smeele (on behalf of all authors)

Reviewer(s)' Comments to Author: Referee 1

I am fully content with the author's responses to my suggestions. Well done on a really interesting and accessible paper.

Again, thank you for your positive feedback on the manuscript.

Reviewer(s)' Comments to Author: Referee 2

I appreciate that the authors have made a thoughtful effort to address all of my previous comments. There are only a few additional points I think should be addressed:

- Imputation
 - A pretty large proportion of life expectancy data (~40%) was imputed – how are the results impacted if these data points are removed.
 - The authors should note that imputed data was all used in the first sentence of Results section.

It is true that a large proportion of the life expectancy variable is imputed. This is, however, the most appropriate way to deal with missing data and results should therefore be more robust than in a model with only complete cases. The third model is subsetted to only include species with known AFR and therefore also only includes three species with unknown life expectancy. This model supports our main findings and we are therefore confident that imputation does not impact the main findings. We included a brief statement in the first sentence of the results ('for all other species life expectancy was imputed in the final models').

- Models & hypotheses
 - The authors made a good effort to clarify the relationships between the different hypotheses and models.
 - However, the evidence required to support/contradict the expensive brain hypothesis (EBH) is still a bit unclear. In particular:
 - The authors state that "If an increase in relative brain size only causes an increase in developmental time (Expensive Brain Hypothesis), we would expect the coefficient of the brain size effect to be positive in model 1 and much reduced or zero in model 2 and 3"
 - And while the coefficient for relative brain size is reduced in models 2 and 3 relative to model 1 (0.22 in model 1, 0.18 in model 2, 0.16 in model 3), they claim no support for the EBH...
 - Exactly how reduced would these estimates need to be to support the EBH?
 - Is the lack of support for the EBH actually due to the non-significant developmental variables? If so, this should be clarified in lines 205- 210.

Thank you very much for the positive feedback and additional suggestions. If the EBH fully explains the correlations, there should be no direct effect of relative brain size, once we control for developmental variables. We made two changes to explain this better: 1) we removed 'much reduced or' from the introduction, 2) we added that we would expect an effect of the developmental variables to the introduction ('We would also expect an effect of developmental variables in model 2 and 3.').